# Industrial Application and Health Prospective of Fig (*Ficus carica*) By-Products

**DOI:** 10.3390/molecules28030960

**Published:** 2023-01-18

**Authors:** Izza Faiz ul Rasool, Afifa Aziz, Waseem Khalid, Hyrije Koraqi, Shahida Anusha Siddiqui, Ammar AL-Farga, Wing-Fu Lai, Anwar Ali

**Affiliations:** 1Department of Food Science, Faculty of Life Sciences, Government College University, Faisalabad 38000, Pakistan; 2Faculty of Food Science and Biotechnology, UBT-Higher Education Institution, St. Rexhep Krasniqi No. 56, 10000 Pristina, Kosovo; 3Campus Straubing for Biotechnology and Sustainability, Technical University of Munich, Essigberg 3, 94315 Straubing, Germany; 4German Institute of Food Technologies (DIL e.V.), Prof.-von-Klitzing Str. 7, 49610 D-Quakenbrück, Germany; 5Department of Biochemistry, College of Sciences, University of Jeddah, Jeddah 21577, Saudi Arabia; 6Department of Urology, Zhejiang Provincial People’s Hospital, Affiliated People’s Hospital, Hangzhou Medical College, Hangzhou 310003, China; 7Department of Food Science and Nutrition, Hong Kong Polytechnic University, Hong Kong, China; 8Department of Epidemiology and Health Statistics, Xiangya School of Public Health, Central South University, Changsha 410017, China

**Keywords:** fig, bioactive compounds, industrial application, health benefits

## Abstract

The current review was carried out on the industrial application of fig by-products and their role against chronic disorders. Fig is basically belonging to fruit and is botanically called *Ficus carica*. There are different parts of fig, including the leaves, fruits, seeds and latex. The fig parts are a rich source of bioactive compounds and phytochemicals including antioxidants, phenolic compounds, polyunsaturated fatty acids, phytosterols and vitamins. These different parts of fig are used in different food industries such as the bakery, dairy and beverage industries. Fig by-products are used in extract or powder form to value the addition of different food products for the purpose of improving the nutritional value and enhancing the stability. Fig by-products are additive-based products which contain high phytochemicals fatty acids, polyphenols and antioxidants. Due to the high bioactive compounds, these products performed a vital role against various diseases including cancer, diabetes, constipation, cardiovascular disease (CVD) and the gastrointestinal tract (GIT). Concussively, fig-based food products may be important for human beings and produce healthy food.

## 1. Introduction

The fig (*Ficus carica*) is an edible fruit of the *Moraceae* family. The fig is commonly grown in an area that extends from Asiatic Europe to North India. However, it is also grown as a crop in most Mediterranean countries due to their warm climates [1]. The Mediterranean region is the area that *F. carica* has steadily migrated to from its origin in western Asia. Worldwide, over a million tons of figs are produced [2]. The common shape of figs is obovoid, turbinate and pear. The color of figs varies according to the variety. Some common hues are creamy, green-yellow, red, copper and florid. The syconium’s walls are thick, pale yellow, golden, pale pink, red or florid. The skin of a fig is fine and thin. About 80% of the overall production of figs took place in Egypt, Israel, Algerian, Iranian, Moroccan, Spain and the United States [3]. The importance of dried fig exchanges is greater. The top producers of dried figs are Turkey (320.0 K metric ton), the United States (27.1 K metric ton), and Spain (59.9 K metric ton) [4]. Furthermore, figs are one of the only five fruit plants mentioned in the Holy Quran [5]. Figs are sweet in taste with high quality fruits and have a high nutritional value [6]. The *Moraceae* family includes the genus *Ficus*, which has over 800 varieties [7]. Figs are a great source of antioxidants and phenolic chemicals [8]. *Ficus carica* is an edible fruit with a substantial commercial value in different forms, including fresh, dried and canned. [9]. It is currently used in many food products such as jams and jellies. The fig fruit has also been used as a filling and garnishing component in baked puddings, cakes and bread. However, biscuits are deficient in several elements, including phenolic compounds, fiber and several of the vitamins and minerals present in fig [10,11]. People continue to utilize fig fruit as a traditional medicine despite its medical benefits being recognized for millennia. Fig contains dietary carbohydrates that have the capacity to postpone digestion, whereas anthocyanins can control hyperglycemia [12]. Consuming dried figs considerably boosts the human plasma antioxidant capability. The number of polyphenols and anthocyanins in fresh figs is related to their overall antioxidant capacity [13]. Free radicals are environmental toxins, including air pollution or cigarette smoke, that affects humans. Antioxidants may help prevent or minimize the cell damage caused by free radicals [14]. The Mediterranean diet is now seen as one which promotes people’s health and wellbeing [15]. The nutritious qualities found in fruits and vegetables are responsible for this effect [16]. These are a great source of phenolic compounds, minerals, amino acids and vitamins [17]. Dried figs have ability to the potentially promote health due to the presence of high levels of polyphenols. [18]. These physiological and nutritional traits are strongly linked to fruit and are typically controlled by the genotype, ripening stage, environmental factors and orchard management techniques [19]. The fig is a nutritious fruit with a large amount of iron and fiber. Eating them as a vegetable may also help individuals to consume enough fiber daily. Fresh or dried fig can be eaten, but people should be aware that dried figs have a higher calorie and sugar content. Because figs are safe to consume, people can utilize them to treat a number of ailments. *Ficus carica* extract has recently been evaluated for its hepatoprotective effects in rats exposed to the hepatotoxic drug rifampicin [20]. However, many of the assertions made in relation to the purported health advantages of figs are not yet supported by sufficient data. The traditional medical systems such as Ayurvedic, Unani and Siddha have all mentioned the therapeutic benefits of *F. carica*. [21]. Fresh plant materials, unprocessed extracts and separated *Ficus carica* components have demonstrated a variety of biological (pharmacological) activity. Fig is valuable against different diseases including disorders of the endocrine (diabetes), respiratory system (liver diseases, asthma and cough), digestive system (ulcer and vomiting) and infectious diseases (skin disease, scabies and gonorrhea) [21]. In the present review, we discussed the consumption of fig and the role of their by-products in food product development against various chronic disorders.

## 2. Bioactive Compounds and Phytochemicals in Fig

Colorful, delicious and nutrient-dense vegetables and fruits are a wonderful addition to our diets. Bioactive substances (polyphenols, carotene, vitamins and anthocyanins) are present in fruits and vegetables which have gained attention due to their potential health benefits [22]. Plants have been an important source of nutrition that has preserved people’s health and improved individuals’ quality of life since the beginning of civilization on Earth [23]. Different bioactive compounds and phytochemicals are shown in Figure 1 and Table 1, respectively.

### 2.1. Vitamins

The two main categories of nutrients are macronutrients and micronutrients. Micronutrients are those that the body needs in tiny amounts, whereas macronutrients are those that the body needs in high amounts. Vitamins and minerals are considered micronutrients that aid humans in developing structural and metabolic processes. *Ficus carica* (fig) contains significant levels of phytochemicals minerals and vitamins and it is prepared in various ways, such as in a fresh, dried, concentrate and paste state. Morgan et al. [43] reported that vitamin C is present in fresh figs from Kadota and Calimyrna. The quantities of vitamin C in fig are near to those of grape and apricots but lower than fresh peach and prunes. The water-soluble vitamin C found in figs is a great natural antioxidant and aids in reducing the non-enzymatic oxidation of vegetables and fruits. It is used to replace artificial antioxidants [44]. The Missions and Adriatic figs contained a minimal quantity of vitamin C. The Black Missions fresh figs have a good source of vitamin A, whereas fresh Kadotas and Calimyrnas contain a low amount of vitamin A. Calimyrnas are essentially light-colored figs. The sulfuring procedure helps Mission and Calimyrnas retain their vitamin A content, but it does not assist the Adriatics variety. Figs have been regarded as one of the healthier fruits with have been linked to a long life, and these fruits also play a significant role in these nations’ traditional Mediterranean cuisine [15]. Antioxidant substances such as polyphenols, organic acids and carotenoids prevent the oxidative processes that may result in degenerative diseases [45].

### 2.2. Antioxidants

Antioxidants can prevent the generation of reactive oxygen substances. Fig extract contains bioactive compounds including total polyphenols, carotenoids, flavonoids and antioxidant properties. Fig extracts from cultivars of a darker color contained more phytochemicals than the extracts of the lighter color variations. Fruit peel contributes the most polyphenols and antioxidant activity compared to fruit pulp [46]. A significant antioxidant capacity in dried *Ficus carica* fruit has been reported. Different fig species’ antioxidant profiles were compared by Abdel-Atyet [38]. *Ficus carica* and *Ficussycomorus* extracts contain several phytochemical compounds that have therapeutic activities, including anticancer properties and anti-inflammatory and antifungal activity. The number of antioxidants in fig extracts varies, making it a great source of healthy natural antioxidants. Some of these variations differ in the amount of solvent used, phenol present or in the interactions between the various extract constituents. An in vitro study showed that dried figs act as antioxidants after being consumed by humans. According to the study, dried fruits make up a larger portion of the food because these are rich in phenol antioxidants and nutrient (fiber) [47]. The common secondary metabolites of plants, known as phenolic compounds, can serve as antioxidants that have good impacts on human health and their physiological activities in plants. Phenolic compounds contain various antioxidants, including quenching singlet oxygen, scavenging free radicals and reducing or donating hydrogen. [48]. Fresh fruits, vegetables, and their products’ colors, flavors and aromas are significantly influenced by phenolic chemicals. In addition to their antioxidative properties, polyphenols may also have anti-carcinogenic, antitumor, anti-inflammatory and antibacterial effects [49]. Figs contain a high level of natural antioxidant substances in which the polyphenol levels are high [47]. Commercially accessible fruits are a good source of antioxidant vitamins that play a meaningful role in the human diet. According to Solomon et al. [46], the antioxidant activity of fig fruits increases with their polyphenol concentration, such as their anthocyanin level. The results showed that fig antioxidants can greatly increase the plasma antioxidant capacity and shield the plasma phospholipid from oxidation [50]. According to Vinson [47], figs are high in sugars, including fructose, dextrose and glucose [51]. Furthermore, the different parts of fig contain various chemicals compounds including biopolymers [13], anthocyanins, polyphenols, fatty acids, several polyphenolic compounds and polysaccharides [52,53]. The different antioxidants properties of fig are show in Figure 2.

### 2.3. Polyphenols

Phenolic substances can play a key role in neutralizing free radicals, the defense of cellular components against hydrogen peroxide and the defense of organs and tissues against peroxide (the degradation of lipids in unsaturated fatty acids). According to Juániz et al. [54], phenolic chemicals are abundant in the *Ficus* species. We consume certain plant-based meals which contain polyphenols. Plants contain lots of antioxidants and may be beneficial for human health. Polyphenols are helpful in preventing or treating cardiovascular disease, diabetes, neurodegenerative illness and problems with weight management [55]. The studies showed that the leaf, pulp and peel contained a biochemical profile, antioxidant properties and anthocyanin with different concentrations [46,56]. Numerous fig accessions have not been compared regarding their phytochemical traits such as the total polyphenolics, carotenoids, antioxidant capacity and particular sugars. Typical plant secondary metabolites, termed phenolics, serve as antioxidants by donating an atom of hydrogen, free radical scavengers and quenching singlet oxygen in addition to serving physiological functions in plants [56]. Figs are a good source of phenolic compounds. In fact, red wine and coffee are understood to be sources of phenolic compounds, but both have a lower phenolic content than fig [18]. In regard to the cultivar, the phenolic content of figs usually varies greatly from one fruit section to another [57]. However, these bioactive components are believed to have no effect on a specific tissue or organ. Due to its bioavailability, a substance can be successfully absorbed from the gastrointestinal tract into the blood and transferred to the right place in the body while remaining bioactive. The results of a previous study proved that humans and animal models can be accurately linked with the measurements of bio accessibility made using in vitro models [58]. The bioactive compounds are extracted from the latex of figs including 6-O-acyl-b-D-glucosyl-bsitosterols or AGS (the acyl methyl group: palmitoyl, linoleyl, stearyl and oleyl). Compared to linoleyl, stearyl and oleyl derivatives, the palmitoyl derivative of AGS functions as the most effective inhibitor for various cancer cell lines [58]. An in vitro study shown that AGS suppresses the growth of the cancer cell lines. The finding showed that AGS is the most effective anticancer agent [59].

### 2.4. Polyunsaturated Fatty Acids

The use of saturated fats, polyunsaturated and monounsaturated fats are regarded as good fats because of their ability to lower the risk of heart disease [14]. Omega-6 fatty acids (31.28%) and omega-3 fatty acids (40.25%) are the two main subgroups of polyunsaturated fatty acids; they are shown in Figure 3. Both of these are crucial healthy fats that the human body requires for both cell development and the brain’s function. However, the human body is unable to produce vital fatty acids. Moreover, these fatty acids can be taken from one’s diet. McCune [60] reported that various areas of a plant or animal contain a different concentration of fatty acids but they do not contain the total levels or proportions in the seeds, flaxseed oil, roots, stems and branches, leaves and blooms tree. Numerous studies have shown that phytosterols lower either the human or animal serum cholesterol levels. The findings showed that fruits, vegetables, seeds and nuts are good sources of phytosterols. The sterols and other biochemical elements discovered in the figs and figs’ leaves which are unsaponifiable have been the focus of a few investigations. The phytosterols present in fig is 433 mg/100 g on a dry basis. Kim et al. [61] found sterol contents in several edible plants, but only three phytosterols were found for figs. (Figure 3).

## 3. Fig and Fig By-product Applications

The amount of consumers who are aware of nutritional and functional foods is increasing. Customers want to consume safe foods that also provide additional benefits alongside its nutritional value. However, customers are more inclined to buy these functional foods due to positive health benefits. Furthermore, people are becoming increasingly interested in functional foods [62]. Numerous functional foods have been created, including biscuits [63], bread [64], meat [65] and probiotics dairy drinks [66]. The characteristics of functional foods are thought to be high levels of protein, fiber and the total phenolic content [67]. The fig, although regarded as a fruit, is actually a flower turned inside out. In figs, the fruit is actually the seeds. It is only fruit that fully matures and becomes semi-dry on the tree. Fig was originally obtained from Asiatic Turkey; it has now migrated from Northern India to all of the nations encircling the Mediterranean. Today, Greece, Turkey, the United States and Spain are the leading manufacturers of dried figs. Fig trees live for a long time and reach fruit-bearing maturity in just two years. Some fig trees in California planted almost a century ago still produce fruit. Calimyrna is an amber-colored variety. Dried figs are available in a variety of forms including as a paste, concentrate, nuggets, powder and diced forms for industrial products. Dried figs can be preserved from yeast fermentation and the growth of mold with the addition of potassium sorbate. Processing increases the moisture content of dried figs. The Calimyrna variety’s color may be stabilized by applying low levels of sulfur dioxide compared to other dried fruits such as apricots or apple [68] Table 2.

### 3.1. Baking Industry

A major portion of the human diet is made up of bakery products because these are prepared from cereal and have a long shelf life. Biscuits are one of the most widely consumed baked goods [79]. Nowadays, most people enjoy consuming bakery products [80]. The world’s rising biscuit consumption is increasing the significance of enhancing cookies [81]. Enriching biscuits with fig seed powder could provide a further pathway for the valorization of food waste. Khapre et al. [73] prepared functional cookies by using fresh Fig (*Ficus carica* L.) fruit powder. The fig powder was incorporated in the preparation of cookies at levels 0, 6, 12 and 18 %. The results showed that fig powder-enriched cookies have a good nutritional value with acceptable organoleptic qualities. Jung et al. [82] investigated the quality characteristics of sourdough bread using fermented fig. The moisture content increased with the increase in the sourdough, whereas the pH value was found to decrease with the increase in the quantity of the sourdough. A higher volume was attained in the 40 % sourdough sample, whereas the baking loss decreased with the increase in the sourdough. The minimum changes in the hunter color were found with the addition of sourdough. The texture and sensory properties of sourdough are different compared to the control.

### 3.2. Beverage Industry

Figs are consumed fresh, and customers seek them due to their excellent flavor. Fresh figs have a sweetness alike to honey, a light acidity and a fruity aroma similar to that of berries and grapes. However, figs are primarily used in the dehydrated form or as a component of jams or other sweet goods because figs are extremely perishable fruits. Figs are affected by post-harvest fungal activity. [83]. There have also been attempts to package fresh fruits to preserve them for the purpose of a longer shelf life. A specific packaging is used because fig is a perishable food [84]. Figs share some characteristics with grapes in terms of their composition and aroma; furthermore, figs are grown in the same area as grapes are grown. Figs have not historically been used as extensively as grapes because of the role grapes play in the production of wine. Some supposedly ancient fig wine recipes have been recorded and preserved [85]. The kinds of yeast utilized in the fermentation process has an impact on the antioxidant content of fermented alcoholic beverages in figs. Therefore, Saccharomyces cerevisiae yeasts have a higher fermentation yield and produce a high concentration in alcoholic beverages. Using non-Saccharomyces yeasts (Pichia fermenting, Wickeromomycesanomalus and Hanseniasporauvarum) in fermentation processes gives a higher concentration and antioxidant capacity than the beverages obtained from *Saccharomy* [86]. Moreover, Ilkin [87] reported that acetic fermentation was carried out to produce certain homemade vinegars based on various recipes including fresh figs, fresh figs that had been dehydrated, fresh figs and apple cider vinegar or fresh figs and grape. Different kinds of raw materials are utilized for the production of fermented alcoholic beverages made from figs. Different products (fresh sliced figs, fig juice and frozen figs) were made by using fermented S. cerevisiae yeast with and without dry fig leaf powder [88]. Numerous Ficus species are used for medical or nutritional purposes in which figs are the most common [89]. Figs have positive effects on human health due to the presence of polyphenols such as flavonoid, rutin and quercetin flavonoid [90,91]. These positive effects are based on their antioxidant capacity [92] and other nutraceutical qualities [93]. These polyphenolic chemicals are present in other fermented beverages as well as beverages made from fruit [94].

### 3.3. Dairy Industry

Yogurt contains probiotics that improve digestion and absorption while increasing the food security in human nutrition to improve individuals’ health [95]. Probiotic-, prebiotic- and symbiotic-based foods are categorized as functional foods. Despite having a high nutritional value, dairy products (including plain yoghurt) are poor suppliers of phenolic compounds and antioxidants. However, cow milk has low levels of phenolic chemicals and fiber [96]. To satisfy the consumer demand for “clean label” foods, vegetables or fruits are added to dairy products to boost their antioxidant benefits and fiber contents [97]. According to Jeong et al. [38], the fig (*Fiscus carica*) is a rich source of carbohydrates, acids, minerals (such as manganese, copper, magnesium and calcium), vitamins (such as vitamin K and β-carotenes), polyphenols, flavonoids, fiber and other substances with healthful properties. This study aims to boost the nutritional value of plain yoghurt by adding figs to increase the phenolics, antioxidants, minerals and fiber contents to fulfil human needs. Therefore, the addition of fig may enhance the yoghurt’s probiotic activity for use in human nutrition and medicine that prevents and treats obesity, hypercholesterolemia, hyper-lipemia, hypertension, gastrointestinal diseases and also encourage the development of gut flora. The fig fruit is used to cure inflammation and paralysis because of its antipyretic, purgative and aphrodisiac qualities [98]. Products made from figs are used to treat a variety of illnesses, including dermatological conditions and atopic dermatitis [99]. Fig extracts are also beneficial for reducing atopic dermatitis symptoms when taken with cortisone. The study showed that fig plant extract has the potential to be employed in the prevention and treatment of skin and cervical tumors [100,101].

## 4. Health Prospective of Fig and Fig-Based By-products

Traditional medical practices have mostly focused on the treatment of dermatological conditions with fig products [102]. Abbasi et al. [103] investigated fig plant extracts’ efficacy in reducing atopic dermatitis symptoms. However, it can also be used in place of corticosteroids. Additionally, another investigation revealed the potential possibility of fig leaf extracts for treating and preventing cervical melanoma [104]. Bergapten and psoralen are two factors of fig leaf extract that have been linked to having anticancer effects. It may be useful in building blocks for creating medicines that inhibit cancer cell proliferation [102]. Furthermore, according to another investigation, figs are useful in cells due to the presence of phenol [105]. Fig extract has shown a potential to be used to develop drugs to treat cardiovascular illnesses because it contains flavone, rutin and quercetin. Additionally, fig leaves and fruits have a significant amount of nutrients and are well-known for having high dietary fiber contents [89]. Fig leaf juice can cure vitiligo due to furanocoumarins, specifically psoralen and daidzein [106] Figure 4, Table 3.

### 4.1. Constipation

Constipation is defined as less than three bowel motions per week [118]. Constipation is assumed to be influenced by psychosocial and behavioral issues including reduced movement, insufficient calorie intake and changes in anorectal feeling. Its multifactorial etiology also includes co-occurring diseases and drug side effects such as pain medications, anticonvulsants, antidepressants and cancer medications [119]. According to the Unani literature, F. *carica* fruit can cure constipation, dysentery, enteritis and piles because it acts as a laxative, purgative, antipyretic and aphrodisiac. Leukoderma and ringworm can also be treated with roots. Constipation is a highly prevalent health issue. However, laxative foods such as figs and their derivatives may be helpful in treating it [120]. Dietary fiber (12.21 g/100 g) is the most popular functional food component that can treat constipation. The typical fig (*Ficus carica* L.) is a broadleaf deciduous shrub belonging to the *Moraceae* family. It is well known for being one of the earliest edible fruits that humans grew in subtropical climates [121]. Different studies have been reported on natural remedies to cure constipation [122]. In the previous study, fresh fig leaves were utilized to treat constipation and similar outcomes were noted in the earlier studies [123]. Furthermore, Ajmal et al. [124] acknowledged the effectiveness of fig leaf extracts in lowering blood sugar levels. However, fig environmental consequences must be reduced and fruit-based goods have a significant benefit as functional additives. Peels and seeds are considered to be less optimal fruits, whereas leaves are a few examples of fruit-based products. Regarding fig by-products, extracts from the peel and leaves may improve the medicinal and nutritious qualities of food items such as sweeteners [125]. The consumption of fig on a regular basis can reduce the incidence of carcinogenesis. Colon cancer can be prevented because the fiber in figs promotes the fast removal of waste from the body. The fig seeds are rich with mucin that binds to waste and fluid in the gut and removes them [126].

### 4.2. Cardiovascular Diseases

Coronary heart disease (CHD) and strokes are examples of cardiovascular diseases (CVDs) that affecte the heart and blood arteries [127]. A specific diet is the most significant modifiable factor to avoid CVDs. There is proof that consuming a plant-based diet reduces the incidence of CVDs [128]. Fruits are the most fundamental ingredients that can help avoid CVDs because they contain polyunsaturated fatty acids [129]. The demand for fig derivatives/by-products (peel, leaves and oil) and fig-based products have increased because they are composed of bioactive components that are beneficial for heart health. Another study determines the phenolic component of fig products that has been impacted by processing (drying and jam preparation) and storage. Fig can reduce the triglyceride levels in humans which can help to maintain heart health. Blood fat molecules called triglycerides are a major contributor to heart disease. Additionally, the antioxidants in figs eliminate free radicals from body which obstruct coronary arteries and lead to cardiovascular disease. According to Soni et al. [130], the fig extract’s ABTS experiment revealed that its antioxidant capacity was quite good. Additionally, figs contain phenols and heart-healthy essential fats and omega-6 fatty acids.

### 4.3. Diabetes

Health professionals are quite concerned about the metabolic condition diabetes mellitus because it is becoming more and more common in both developed and developing nations. In 1985, diabetes was declared to affect 30 million individuals according to the World Health Organization. According to the WHO [131], that number rose to 135 million in 1995, and 300 million people are anticipated to be impacted by the year 2025. Worldwide, there is a health concern regarding the attempt to manage diabetes and its consequences. Plant-based medicinal medications for the treatment of diabetic diseases have received a lot of attention in recent years due to its accessibility, affordability and lack of adverse side effects. Different cultures around the world have used a variety of medicinal plants in traditional medicine to stop long-term issues in managing diabetes. About 200 uncontaminated bioactive components have been extracted from therapeutic plants including polyphenols, triterpenoids and starches. These compounds have powerful antidiabetic activities because they control blood glucose levels [132]. The *Moraceae* family contains approximately 850 plants, including trees and shrubs, vines, epiphytes and the genus *Ficus*. These trees, bushes, plants and epiphytes are found throughout humid and subtropical regions of Southeast Asia, tropical America and Australia, where there is the highest variety [133]. In Siddha, Ayurvedic and Chinese traditional medicine, multiple *Ficus* species are utilized for a variety of therapeutic purposes [85]. Originally from Asia Minor, F. *carica* is a fruit shrub that is common in tropical and subtropical regions. Plants can treat different diseases including diabetes, liver disorders, cough, ulcers, nausea, menstrual cramps, crust conditions and gonorrhea [21]. In Pakistan, leaf decoction is also used to manage diabetes [134]. The *Ficus* species have important hypoglycemic properties which include improving insulin sensitivity, causing the release of insulin, increasing hepatocellular aerobic glycolysis, decreasing the up-take of carbs, attempting to regulate bowel tract enzyme reactions, increasing the peripheral glucose uptake and boosting the antioxidant status. A previous study showed that the *Ficus* species are potent in anti-diabetic medications. The in vitro and in vivo studies showed that the essential oil and polyphenol components from many *Ficus* species have anti-diabetic properties. All diabetes-related problems were significantly reduced by the chemicals extracted from Ficus species in streptozotocin and allowance-induced diabetic mice. The most important antidiabetic effects of the Ficus species have been demonstrated, including an increased insulin awareness, an increase in insulin secretion, encouraged hepatic glycogen synthesis, reduced carbohydrate absorption, controlled intestinal enzyme activities, enhanced hepatocyte gluconeogenesis and enhanced antioxidant potential [135].

## 5. Conclusions

It is concluded that fig is a rich source of nutrients and bioactive compounds. Different parts of fig (the leaves, seeds and latex) are contained in different phytochemicals. The change in the different concentrations varies based on the part of the fig. Due to the rich source of phytochemicals, fig by-products play an additive role in the development of different food products, including beverage, dairy and bakery industries. Because these products are enriched with bioactive compounds, they are considered to be suitable for various disorders. In the future, fig-based nanoparticles can be used as a coating material for a different food.

## Figures and Tables

**Figure 1 molecules-28-00960-f001:**
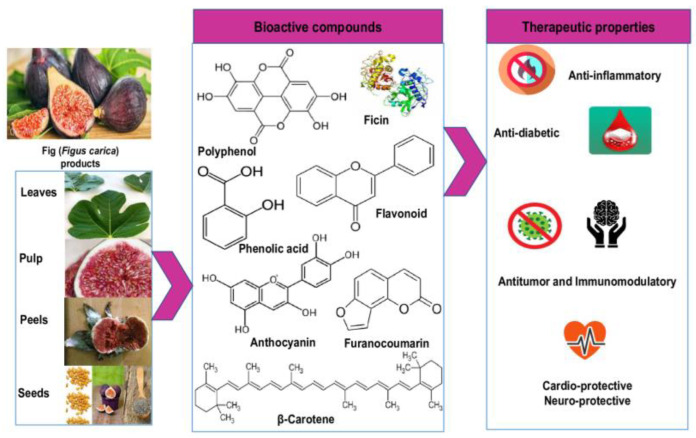
Bioactive compounds and biological properties of fig.

**Figure 2 molecules-28-00960-f002:**
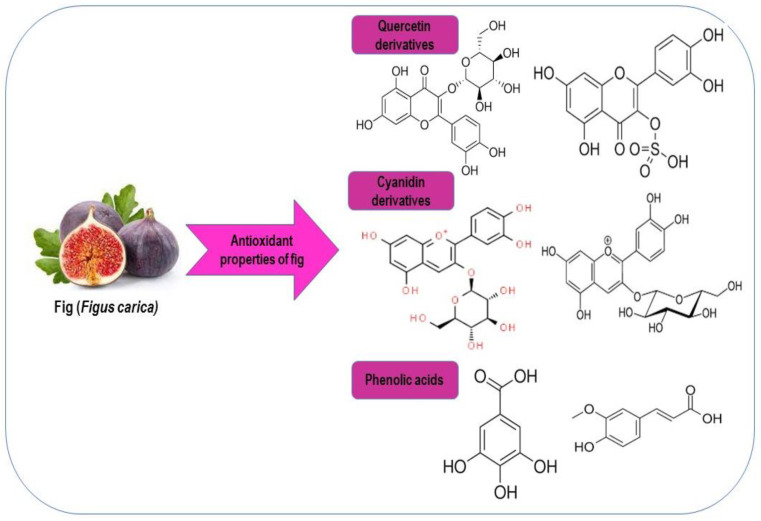
Antioxidants properties of fig.

**Figure 3 molecules-28-00960-f003:**
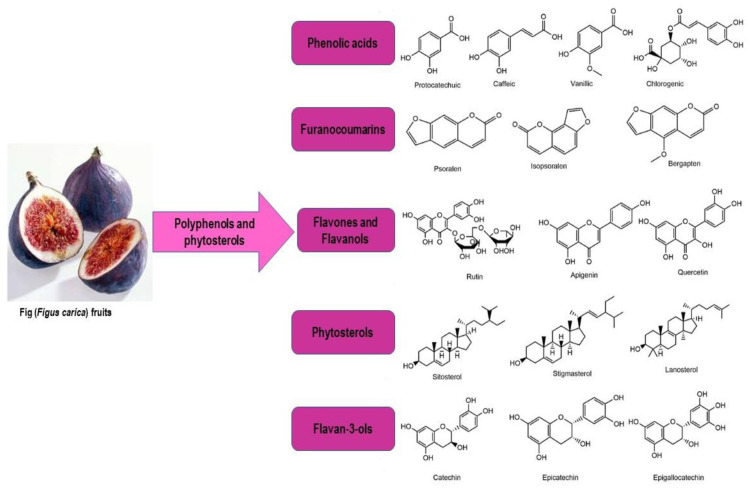
Polyphenols and phytosterols of fig.

**Figure 4 molecules-28-00960-f004:**
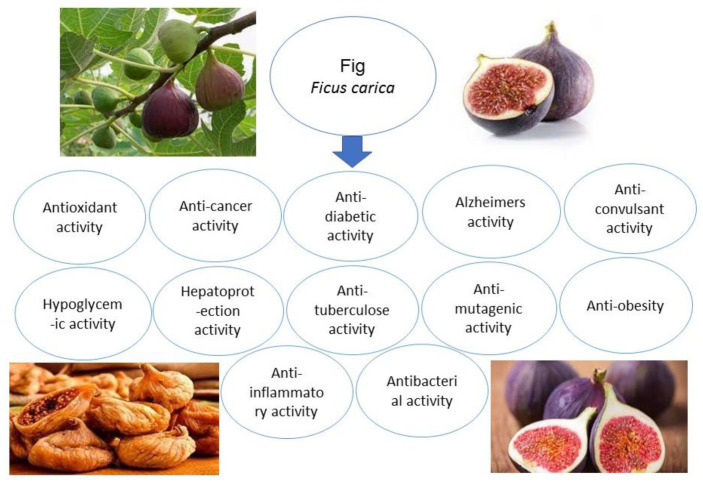
Health benefits of fig.

**Table 1 molecules-28-00960-t001:** Phytochemical Composition in different parts of fig.

Different Parts	Extraction Method	Phytochemicals	Solvent	References
Fermented fig by-product	High-pressure assisted extraction	Antioxidants, total phenolic, tannin and flavonoid	Ethanol	[24]
Fig leaves	Tailor-made deep eutectic solvent (DES) and microwave-assisted extraction	Caffeoylmalic acid, psoralic acid-glucoside, rutin, psoralen and bergapten	Methanol	[25]
Whole fig fruit, peel, leaves and pulp	--	Total phenols and antioxidant	--	[26]
Ficus carica L. latex	Maceration and ultrasound-assisted extraction (UAE)	Total phenolic content (TPC) and antioxidant	Methanol, ethanol, ethyl acetate and n-hexane	[27]
Fig leaves	Surfactant-based microwave-assisted extraction	Caffeoylmalic acid, psoralic acid-glucoside, rutin, psoralen and bergapten reached	Ethanol	[28]
Fruit, leaves and stembark	Ethanolic extract	Butyl butyrate, 5-hydroxymethyl furfural, 1-butoxy-1-isobutoxy butane, malic acid, tetradecanoic acid, phytol acetate, trans phytol, n-hexadecanoic acid, 9Z,12Z-octadecadienoic acid, stearic acid, sitosterol, 3,5-dihydroxy-6-methyl-2,3-dihydro-4H-pyran-4-one and 2,4,5-trimethyl-2,4-dihydro-3H-pyrazol-3-one	Hexane, ethyl acetate, ethanol and water	[29]
Ficus carica L. peel	Heat, microwave and ultrasound	Anthocyanin	Ethanol and water	[30]
*Ficus carica*	Ethyl acetate	Phenol groups and amine groups	--	[31]
Leaves	Soxhlet extractor	Quercetin, chlorogenic acid, caffeic acid, syringic acid, coumaric acid, rutin and trans-cinnamic acid	Water, methanol and ethanol	[32]
Fig leaf	Ethanolic	Antioxidant	Ethanol	[33]
Fruit	Hexane extract	Chemical compounds and antibacterial	Hexane	[34]
*Ficus carica*	Methanol (MeOH) extract	Antimicrobial	Methanol	[35]
Black fresh fig	Solvent extraction	Antioxidants and total phenolic contents	Acetone	[36]
Leaf	Methanol and water extracts	α-glucosidase and α-amylase	Water	[37]
Latex	Methanol extracts	Total phenolic and flavonoid contents	Methanol	[38]
Fruit	Ethanol and the temperature of extraction	Total phenolics, total flavonoids and total proanthocyanidins	Ethanol	[39]
Fresh fig	Methanol water extraction	Total phenolics and total anthocyanins	Methanol and water	[40]
Seed	Solvent extraction	Polyphenol contents and antioxidant	Acetone, methanol and ethanol	[41]
Leaves	Solvent extraction	Biological active lipid components	Diethyl ether, petroleum ether, n-hexane, acetone and ethanol	[42]

**Table 2 molecules-28-00960-t002:** Value addition role of fig in food industries.

Industry	Food Product	Additive	Function	Reference
Food industry	Fig powder co-products (FPC)	Peel and pulp	FPC obtained from peel present higher antioxidant activity than FPC obtained from pulp.	[69]
Food industry	Functional food	Leaves, pulp, peels, seeds and latex	It is the high value-added ingredients and their utilization in novel food formulation development.	[70]
Food packaging	Fig-based Chitosan film	Leaves	Fig leaves extract incorporated chitosan films can be used for protection of the food items and increase their shelf life.	[71]
Confectionery	Doughnut icing and pastry product “beijinhos”	Fig peels	Use of fig peels fruits as natural colorants.	[72]
Baking	Cookies	Fresh fig fruit powder	Fig powder-incorporated cookies have rich nutrients as compared to market products.	[73]
Snack	Fruit-based snack	Dried fig	In this study, a novel snack based on fig fruit powder was developed.	[74]
Dairy	Sugar-free milk-based dessert	Dried figs and CMC	Figs and CMC influenced the dessert’s characteristics.	[75]
Oil	Canola oil	Pulp and skin extract	The high efficiency of fig skin extract for oxidative stability is assessed because it is good source of phenolic compounds.	[76]
Bakery	Burfi (Indian cookie)	Fig fruit powder	The fig fruit powder-based product was found to be low cost as compared to market products.	[77]
Confectionery	Toffee	Fig fruit powder	The toffee products prepared by fig fruit powder were assessed for their physico-chemical and sensory parameters.	[78]

**Table 3 molecules-28-00960-t003:** Role of different by-product of fig against chronic disorders.

Fig Part	In vivo/In Vitro	Disease	Action	Reference
Leaves and fruit	In vivo and in vitro	Inflammatory and Carcinogenic effects	In vivo and in vitro study illuminates that F. *carica* leaves and fruits play important role the prevention of inflammatory and carcinogenic effects.	[107]
Leaves	In vivo	Kidney	The findings showed that fig extract play important role in the treatment of kidney diseases.	[108]
*Ficus carica*	---	Anemia, cancer, diabetes, leprosy, liver diseases, paralysis and ulcers	*Ficus carica* is a good source of traditional medicine for the treatment of anemia, cancer, diabetes, leprosy, liver diseases, paralysis and ulcers.	[21]
*Ficus carica* leaf extracts	In vitro	Cancer, diabetes mellitus and Alzheimer’s disease	The results suggest that F. *carica* leaves may be valuable source for developing a promising therapeutic agent in cancer, diabetes and Alzheimer’s disease.	[37]
*Ficus carica* polysaccharides (FCPS)	In vivo	Aeromonas hydrophila infection	Results showed that dietary FCPS can be improved the innate immune response, growth performance and disease resistance against A. hydrophila in fish.	[109]
*Ficus carica* stem extract	In vivo	Hepatic oxidative damage	Fig extract played significant role in protecting animals from methanol-induced hepatic oxidative damage.	[110]
*Ficus carica* cell suspension culture extract	In vitro (on keratinocytes cells) and in vivo	Skin	In vitro and in vivo tests demonstrated that the extract from *Ficus carica* cell cultures alleviated skin damage caused by psychological stress.	[111]
Acetonic extract of *Ficus carica* L.	In vivo	Central nervous system	The outcomes showed that *Ficus carica* L. can be useful in insomnia, anxiety, schizophrenia, migraine and epilepsy.	[112]
*Ficus carica* (fig) and Olea europaea (olive)	--	Pro-inflammatory cytokines	Consuming figs and olives can be useful in the prevention or treatment of inflammatory diseases.	[113]
Leaves of *Ficus carica*	In vivo	Hepatotoxicity	Fig leaves extract reduce the risk of hepatotoxicity.	[114]
Leaf and fruit	In vitro	Cancer	Results suggest that leaf and fruit have anticancer activity.	[115]
Leaves	In vivo	Diabetes	Reduce the risk of diabetes.	[116]
Leaf and fruit	--	Hyperglycemia	The fig fruit or its leaves in food may help to correct the hyperglycemia due to diabetes.	[117]

## Data Availability

Not available.

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
