# Peer review of "Industrial Application and Health Prospective of Fig (Ficus carica) By-Products"

_molecules, 2023, doi:10.3390/molecules28030960_

Round 1

Reviewer 1 Report

Add “byproducts” after “fig” in title

Line 29 and 30, delete “industry” replace with industries at the end of sentence as “diary, bakery and beverage industries

Line 30 fig byproducts use also in line 31

Indicate the abbreviations in abstract

Enrich the abstract

Correct linguistic errors in the manuscript

Line 61 incomplete sentence

Lines 74 and 82 it should be Ficus carica

Clear the objectives of review

The authors didn’t clear in introduction “Fig or Fig byproducts” , if they talk about figs where the novelty

Use American English

Redesign Figure 1

Add full profile for Fig and Fig parts using LC-MS

In Table 1 columns, the authors should correct “antioxidant” to “Antioxidant products such as “…”

In the last two column clear what is the solvent used in extraction

The authors use the same title in sec 2 and 3 it should be in sec 3 as Fig and Fig by products applications

Rewrite sec 3.1

Line 280, add β before carotenes or delete “-“

Table 3, column “recovery” replaces with “action” and enhance the expression of those actions

Line 400, different concentrations

Scientific names must be italic

Check the outputs of all references

Author Response

Add “byproducts” after “fig” in title

Response: Added 

Line 29 and 30, delete “industry” replace with industries at the end of sentence as “diary, bakery and beverage industries

Response: Replaced 

Line 30 fig byproducts use also in line 31

Response: added

Indicate the abbreviations in abstract

Response: Added

Enrich the abstract

Response: Enriched

Correct linguistic errors in the manuscript

Response: Corrections have been made in the revised version

Line 61 incomplete sentence

Response: The sentence has been completed 

Lines 74 and 82 it should be Ficus carica

Response: Changes have been made

Clear the objectives of review

Response: Added

The authors didn’t clear in introduction “Fig or Fig byproducts” , if they talk about figs where the novelty

Response: Chances have been made

Use American English

Response: done 

Redesign Figure 1

Response: Figure has been redesigned according to suggestion of reviewer

Add full profile for Fig and Fig parts using LC-MS

Response: added

In Table 1 columns, the authors should correct “antioxidant” to “Antioxidant products such as “…”

In the last two column clear what is the solvent used in extraction

Response: The solvents have been added according to suggestion of reviewer.  

The authors use the same title in sec 2 and 3 it should be in sec 3 as Fig and Fig by products applications

Response: added 

Rewrite sec 3.1

Response: The section has been rewritten 

Line 280, add β before carotenes or delete “-“

Response: added

Table 3, column “recovery” replaces with “action” and enhance the expression of those actions

Response: Action take according to suggestion of reviewer

Line 400, different concentrations

Response: added 

Scientific names must be italic

 Response: done

Check the outputs of all references

Response: checked 

Reviewer 2 Report

Dear authors,

your manuscript is very interesting. You have described important parts, composition and uses of the fig. It is my opinion that the manuscript is well divided into subsections and every section contains relevant information and references. However, I have several recommendation for improvement:

- First, English changes are required; there some grammatical errors, missing verbs etc.

- Line 17-23: please change numbers 5-9 into superscript.

- Line 39: The first sentence of the Introduction is missing a verb and therefore it is harder to understand the point of it.

- Line 65: I believe that "what are" is not necessary.

- Line 88-89: This sentence repeats what is said in the previous sentence.

- Tables: Could they be organized differently in order to fit a page or to be easier to read (e.g. the last column with references doesn't have to be so wide).

- Line 74: "Ficuscarica" should be two words. Check the text, there are several same errors.

- Is there a possibility to enlarge Figure 3, or to enlarge the font of the compounds name?

- Line 202: Subtitle no. 3 is same as no. 2?

- In reference list there is 136 references, but in text only 135, and I believe that references 134 and 135 are not suitable for the text where they are stated. Please check references.

After these changes, it is my opinion that the manuscript could be considered to be published in this Special Issue.

Author Response

Dear authors,

your manuscript is very interesting. You have described important parts, composition and uses of the fig. It is my opinion that the manuscript is well divided into subsections and every section contains relevant information and references. However, I have several recommendation for improvement:

Response: We appreciate the reviewer for the relevant information.

- First, English changes are required; there some grammatical errors, missing verbs etc.

Response: The article has been edited by the senior colleague.

- Line 17-23: please change numbers 5-9 into superscript.

Response: changes have been made

- Line 39: The first sentence of the Introduction is missing a verb and therefore it is harder to understand the point of it.

Response: changes have been made

- Line 65: I believe that "what are" is not necessary.

Response: removed

- Line 88-89: This sentence repeats what is said in the previous sentence.

Response: The sentence has been deleted according to suggestion of reviewer

- Tables: Could they be organized differently in order to fit a page or to be easier to read (e.g. the last column with references doesn't have to be so wide).

Response: Yes, Changes have been made

- Line 74: "Ficuscarica" should be two words. Check the text, there are several same errors.

Response: Corrected

- Is there a possibility to enlarge Figure 3, or to enlarge the font of the compounds name?

Response: yes, enlarged 

- Line 202: Subtitle no. 3 is same as no. 2?

Response: no, It was just by mistake. However, heading has been added. 

- In reference list there is 136 references, but in text only 135, and I believe that references 134 and 135 are not suitable for the text where they are stated. Please check references.

Response: Corrections have been made

After these changes, it is my opinion that the manuscript could be considered to be published in this Special Issue.

Response: we are thankful to reviewer for the valuable comments 

Reviewer 3 Report

Dear Authors,

the subject of the work concerns the interesting issue of using fig in the diet and food production, detailing the numerous health-promoting properties of various parts of the plant.

I present the remarks to the manuscript in the file.

Kind regards

Reviewer

Author Response

Dear Authors,

the subject of the work concerns the interesting issue of using fig in the diet and food production, detailing the numerous health-promoting properties of various parts of the plant.

Response: The authors are highly thankful to the respected reviewer for kind suggestions/comments for improving the manuscript. We tried to revise the manuscript according to the valuable advices.

I present the remarks to the manuscript in the file.

Response: We carefully revised the whole manuscript including your suggestions and comments.

Round 2

Reviewer 1 Report

Now can be accepted in Molecules